# Integration of Blockchain Technology and Federated Learning in Vehicular (IoT) Networks: A Comprehensive Survey

**DOI:** 10.3390/s22124394

**Published:** 2022-06-10

**Authors:** Abdul Rehman Javed, Muhammad Abul Hassan, Faisal Shahzad, Waqas Ahmed, Saurabh Singh, Thar Baker, Thippa Reddy Gadekallu

**Affiliations:** 1Department of Cyber Security, Air University Islamabad, Islamabad 44000, Pakistan; abdulrehman.cs@au.edu.pk (A.R.J.); faisal.rwp@gmail.com (F.S.); waqaskhattak99@gmail.com (W.A.); 2Department of Computing and Technology, Abasyn University, Peshawar 25000, Pakistan; abulhassan900@gmail.com; 3Department of Industrial and System Engineering, Dongguk University, Seoul 04620, Korea; saurabh89@dongguk.edu; 4Department of Computer Science, College of Computing and Informatics, University of Sharjah, Sharjah P.O. Box 27272, United Arab Emirates; tshamsa@sharjah.ac.ae; 5School of Information Technology and Engineering, Vellore Institute of Technology, Vellore 632014, India

**Keywords:** blockchain, federated learning, intelligence transportation system, vehicular internet of things (IoT), vehicular ad hoc network (VANET)

## Abstract

The Internet of Things (IoT) revitalizes the world with tremendous capabilities and potential to be utilized in vehicular networks. The Smart Transport Infrastructure (STI) era depends mainly on the IoT. Advanced machine learning (ML) techniques are being used to strengthen the STI smartness further. However, some decisions are very challenging due to the vast number of STI components and big data generated from STIs. Computation cost, communication overheads, and privacy issues are significant concerns for wide-scale ML adoption within STI. These issues can be addressed using Federated Learning (FL) and blockchain. FL can be used to address the issues of privacy preservation and handling big data generated in STI management and control. Blockchain is a distributed ledger that can store data while providing trust and integrity assurance. Blockchain can be a solution to data integrity and can add more security to the STI. This survey initially explores the vehicular network and STI in detail and sheds light on the blockchain and FL with real-world implementations. Then, FL and blockchain applications in the Vehicular Ad Hoc Network (VANET) environment from security and privacy perspectives are discussed in detail. In the end, the paper focuses on the current research challenges and future research directions related to integrating FL and blockchain for vehicular networks.

## 1. Introduction

The Internet of Things (IoT) usage is increasing in both industrial and consumer applications. According to a previous estimation, around 50 billion IoT devices have been installed by 2020 [1]. Aside from the healthcare business, one of the key areas where IoT is widely employed is in-vehicle networks. Vehicular networks evolved from initial Vehicular Ad-hoc Networks (VANETs) to Vehicular Sensor Networks (VSN), which are composed of a connected, scalable, talk-able, and sense-able mesh of devices in which vehicles are a key component of the overall Intelligent Transport Infrastructure (ITI) [2,3,4,5].

ITI emphasizes (i) improving commuter safety, (ii) vehicle communication, (iii) traffic management, and (iv) intelligent decision making. The ITI is split into two sections: infrastructure-oriented and infrastructure-free. The infrastructure-oriented is permanently installed and primarily monitors and controls components such as Road Side Units (RSUs) and infrastructure-free consists of participant automobiles [6,7].

Security is the most important necessity of such real-time systems, because a single mistake or problem might result in death. All three pillars of security must be addressed in such systems: (i) Confidentiality, (ii) Integrity, and (iii) Availability. The ITI’s network is the most important point of entry. There are three main domains, which are (i) Ad-hoc Network Domain (AHND), (ii) Road Side Network Domain (RSND), and (iii) In-Vehicle Network Domain (IVND). The paramount communication flows among these domains are vehicle to vehicle (V2V) and RSU (V2R). Data transmission across various domains and information are given via the communication channel are critical [8]. Data exchange across various domains and information transmitted through the communication channel are critical in vehicular networks [8]. The VANET’s security is directly proportional to the number of reliable components providing accurate and timely data through the network. Major attacks on these domains are likely to include but are not limited to data sniffing, spoofing with malicious intent, malicious data injection, and assaults on data or network availability. In the majority of the cases, one can identify that the attacks are related to data shared on the vehicular network [9]. To ensure a safe and trusted ITI, data integrity, privacy, and confidentiality assurance are the main goals.

Blockchain and Federated Learning (FL) are two emerging technologies that may be employed in ITI to solve privacy and security concerns while also limiting communication costs and latency problems. Blockchain is a digital ledger for record-keeping [10,11]. It is an decentralized and dispersed in nature with tamper-resistant and tamper-evident features [12,13,14]. Google Inc. introduced the FL idea to solve privacy preservation and communication overhead Issues that emerged from combining data from several nodes and storing it in a centralized location [15,16]. Machine Learning (ML) and Deep Learning (DL) algorithms provide better outcomes if more data is allocated for the model’s training; nevertheless, processing such a large quantity of data takes longer during training. Training with fewer data takes less time and yields a lower accuracy score, according to [17]. Blockchain may provide vehicle integrity and trustworthiness, FL can be used to preserve privacy, and encryption for confidentiality ensures realistic and implementable solutions that can be implemented into the vehicular network. Motivated by these, we have attempted to provide a comprehensive survey on integrating blockchain and FL in vehicular networks in this work.

### 1.1. Comparison and Contribution

Several researchers have separately presented engaging surveys on FL for vehicular networks [16,18,19,20,21] and blockchain for vehicular networks [22,23,24,25,26]. However, to the best of our knowledge, this is the first time that blockchain and FL have been combined for vehicle networks. Previously researchers examined IoT and blockchain applications in vehicular networks [22,27,28,29,30,31,32,33,34]. However, those surveys are not concerned with blockchain and FL applications in IoV networks. On the other hand, recently published research papers mainly focused on the blockchain applications in IoV networks [35,36,37,38,39,40,41,42]. Therefore, comparing the proposed survey with the existing surveys, our proposed survey focuses on integrating blockchain and FL in the IoV environment from security, privacy, and energy efficiency perspectives.

The main contributions of this survey are summarized as follows:Elaborates VIoTs, blockchain, FL, and intelligent transportation infrastructure;Integrates blockchain and FL in a VIoT context focusing on privacy and security;Demonstrates various recent VIoT-related real-world projects;Addresses the role of state-of-the-art STI in VANET, such as vehicular networks, roadside infrastructure, and smart automobiles;Provides future blockchain and FL research in the VIoT space.

### 1.2. Survey Structure

Section 1 presents the detailed introduction of the proposed survey that includes survey contributions and structure. Section 2 presents the state-of-the-art smart transport infrastructure from a VANET environment perspective that includes smart vehicles, roadside infrastructure, vehicular network, support decision networks, sensors and actuators, federated STI, and ML and DL concepts. Section 3 presents FL, distributed learning, and FL integration in VANET. Section 4 presents blockchain and its integration with FL in VANET. Section 5 presents the existing applications of FL and blockchain in the VANET environment from a security and privacy perspective. Section 6 presents real-world projects associated with the VANET environment from a security perspective. Section 7 presents research challenges regarding FL and blockchain integration along with possible future research direction from the FL and blockchain perspectives in the VANET environment. Section 8 summarizes the proposed survey. Figure 1 depicts the context of the proposed survey.

## 2. State-of-the-Art Smart Transport Infrastructure

STI has become a crucial component of almost every modern city, smart or conventional. The traffic load in modern cities is heavy and diverse, requiring specific attention and planning to manage it more efficiently and effectively. Vehicular traffic management is a fundamental need and must be managed and regulated to ease inhabitants’ daily lives. STI must be capable of accurate detection, designed for high throughput real-time processes, and designed for lightweight systems to incorporate IoT [43]. As discussed in the published literature, the significant issues revolve around security, privacy preservation, protection, and energy-efficient utilization [44]. Significant components of the STI are detailed in the following subsections. Figure 2 presents the infrastructure of the smart transport system in a vehicular network.

### 2.1. Smart Vehicles

Smart car development is in full swing by major car manufacturing companies, and international competition exists in this field. Major Chinese and US-based technology companies, such as Xiaomi, Huawei, Baidu, Alibaba, and Apple, are joining the electric car manufacturing companies for the production of smart cars [45,46]. The promising feature of smart vehicles is their self-driving capability. The significant components of such vehicles are fully connected, having an Advanced Driver Assistance System (ADAS) [47], Adaptive Cruise Control (ACC), Alcohol Ignition interlock device (AIID), Antilock Braking System (ABS), Collision avoidance system (CAS), Blind Spots Monitoring (BSM), Electronic Stability Control (ESC), Driver Monitoring System (DMS), camera and vision control, Automatically Commanded Steering Function (ACSF), omniview 360, Traffic Sign Recognition (TSR), communication control manager, and data processing units. The most critical challenge for such components is to ensure security while attaining real-time data processing and communication from/to various attributes/components of the overall system. Energy efficiency is a relatively minor issue as the vehicle’s power source power the vehicle components [48].

### 2.2. Road Side Infrastructure

Roadside Infrastructure (RSI) is an important aspect of the entire STI. It comprises vehicle counters, speed cameras, monitoring cameras, accident prediction systems, infrared sensors, vehicle flow managers, and advanced computer vision for tracking cars. To effectively and efficiently operate the overall system, these systems communicate vital information to other systems and vehicles passing through them. The constraints for those systems include security, privacy protection, and energy-efficient operations of the components that reside within the RSU domain.

### 2.3. Vehicular Network

Machines are meant to do specific tasks in a distributed mechanism [49]. In such phenomena, immense duties may be accomplished faster by employing more machines instead of a reduced number of machines. Optimization is accomplished by adjusting the ratio of the advantages acquired from early completion in relation to the cost of machine operations. Machines were initially designed to be static and serve only a single function. Afterward, advancements were made to these machines in the second generation, allowing machines to tackle comparable tasks with a single instance. Electronic advancements and the advent of chips have greatly enhanced the current lifestyle, and the third generation of machines has increased control, accuracy, and automation. Design, awareness, and intelligence are all significantly improved in fourth-generation machines. 3D printing, nanotechnology, and artificial intelligence make it easier for people to be replaced by machines in activities that demand much effort and/or are conducted in hazardous situations. The interconnected, context-aware, decision-making intelligent machines that will drive the progress into the fifth generation are the emerging trend in this sector [50,51]. Figure 3 shows the overview of the evolution of technology in machines.

Decision support systems operate as the brain of the STI. They acquire data from many sources, including automobiles, roadside sensors, roadside monitors, smart cameras, and weather sensors. One of the systems exploiting the powers and empowerment of the fifth-generation machine is the smart vehicular network. In the situation of smart cities, smart transportation is the critical aspect of the total infrastructure [52]. The smart transportation infrastructure is a complex mesh of network devices, smart cars, monitoring systems, sensors, image and vision recognition, peer-to-peer communication, and risk management techniques to accomplish a shared goal.

The smart vehicles of the modern day have connectivity and compute engines that process the data and transmit/receive information inside and outside the vehicle. The On-Board Computer (OBC) interconnects major components/systems inside the vehicle and manages information exchanges. However, the foremost impacting ability of today’s smart cars is to communicate and coordinate with vehicles and other road infrastructure components surrounding them. The new genre of vehicles is smart and linked [53]. This smartness and connected approach eliminate many concerns and obstacles in advancing towards completely autonomous smart vehicle operations, yet they confront numerous open challenges. These challenges include unstable network nodes (vehicles), high throughput requirements to be real-time operations, the common language of communication among vehicles from different vendors, privacy and security challenges, and integrity of the data being shared over these networks [54].

### 2.4. Decision Support System

Decision support systems play a pivotal role in the overall system safety and stability. Data processed from different sources are analyzed against a framework or model for the identification of possible hazards [55]. The discovered vulnerabilities are then conveyed in real-time to other participant systems in STI for information, and appropriate measures are started to limit the hazard’s severity before completing the same cycle again with fresh data. A city-level decision support system for STI is broken into separate chunks, and every chunk evaluates and detects threats to its immediate area and communicates the same information to the central system. The central system regularly evaluates the data from all chunks, updates its central model for city-level projection of concerns and risks, and updates Law Enforcement Agencies (LEAs) and emergency services for assistance if necessary. Significant challenges to these systems are the integrity of received data, processing and computation overhead, time limitations being a real-time system, and privacy preservation [56].

### 2.5. Sensors and Actuators

Sensors and actuators in STI are divided into many categories. There are in-vehicle sensors that monitor different aspects of the vehicle like speed, temperature, the status of different internal systems, power, engine oil, transmission, propulsion, radar, laser range finders, navigation, and obstacles [57] as well as externally placed in RSUs for monitoring the infrastructure and vehicles traveling on these infrastructures like traffic flow monitoring, accident tracking, automated parking systems, context awareness, and safety and security measures monitoring [58,59].

### 2.6. Federated Smart Transport Infrastructure

FL has evolved as a potential solution to privacy problems. FL’s goal is to prevent local data from being shared [53]. Federated Smart Transport Infrastructure (FSTI) is a novel research area that is being actively investigated under the domain names Federated Vehicular Network (FVN), FL in VANETs, ML for Vehicular IoT networks, or STI, Intelligent Transport System (ITS), and Smart VANETs. The open challenges that FSTI is facing are lowering the communication overhead in real-time systems, faster computation and efficient processing, defining a standard architecture and boundary for local and central data, and preserving the power usage while keeping in view the limited batteries of different components within the system, and network advancements/enhancements.

### 2.7. Machine Learning and Deep Learning

THE ML-based VANET system needs sensitive information to choose diverse aspects, such as selection of training and communication. Nowadays, with the help of privacy mechanisms, most sensitive information is disguised. Furthermore, On-Board Units (OBUs) require too many resources in the VANET environment, and consequently, ML-based and misuse-based learning approaches are not suited for vehicles network. For the VANET environment, a lightweight, sensitive information transfer protocol is necessary for VANET [60].

From the resource requirement standpoint, high-performance computational resources must train DL-based models [61]. DL has been extensively involved and gained important attention in several domains of VANET to enhance the performance of various tasks, such as anomaly detection in vehicular communication, network traffic classification, and attack detection [3,62]. DL models automatically evaluate and extract useful features from the supplied data. In a VANET context, DL-based models achieve high performance based on the raw traffic inputs and with minimum resources, and this is a suitable technique that can be applied in a vehicular network.

DL has been widely involved and gained significant consideration in different fields of VANET to improve the performance of various tasks, such as anomaly detection in vehicular communication, network traffic classification, and attack detection [3,62].

In the VANET environment, integrating intervehicle communications with ML and DL can achieve different applications and safety procedures such as secure communication, intrusion detection, prediction of wide traffic congestion, predicting the collision, object detection, early anomaly detection and prevention, smart unmanned vehicle management, smart fleet monitoring, identification, and cross border traffic management [3,7,62,63,64,65,66,67]. Explicitly, with the support of DL, the environmental perception is accomplished using existing vehicle sensor sensitive data. With the help of exchanging traffic-associated information, the extended perception range is achieved via communication between vehicles. Intelligent Vehicles (IV) increase their self-observation range via the information distribution techniques and, with the help of self-collected data, percept the neighboring environment in a VANET. However, numerous privacy and security challenges exist in the VANET environment because of the dynamic changes, such as resource management, data transmission, packet loss in real-time scenarios, and computation power.

## 3. Background

### 3.1. Distributed Learning

For the training of complex ML applications, a significant amount of data is required. As the number of parameters rises throughout the data training process, the data input in larger ML models likewise increases. Moreover, since the need for processing training data has risen regarding computing resources across several machines, distributing ML has become essential [68]. Some computing architectures and paradigms have been presented for the solution (i.e., peer-to-peer, Apache Spark, all-reduce, parameter server, and TensorFlow [53]).

### 3.2. Federated Learning

FL has been considered in critical networks to handle the centralized data problems (e.g., VANET) as a solution. It is a sort of distributed ML model with benefits of AI-based end-devices computation applications, such as smart cars, and also guarantees the privacy of end-users [69,70]. The ML/DL model is trained globally and distributes the updated parameters to end devices’ via the centralized server to start the process in FL. With the assistance of this trained model, all end devices train their local model of ML/DL on their local data. Clients send the parameters to a centralized server for global training upon training the local ML model. To fulfill the demand and precision of centralized servers, the operation is repeated multiple times until it is all done. There are several applications, such as e-commerce and e-healthcare, where FL can be conveniently deployed [69,71], as is shown in Figure 4.

In ML, the accuracy and efficiency of trained models rely on the central server’s training data and processing resources (Computational Power). Throughout the conventional ML approach, centrally stored data are utilized for both training and testing purposes to construct efficient and comprehensive models. Several challenges are associated with centralized ML algorithms concerning user data, including time, computational power, privacy, and security. To solve the described concerns as a technology solution, FL has recently emerged [70]. The FL approach incorporates statistical model training over data centers or remote devices, e.g., hospitals and mobile phones, keeping user data localized. Statistical model training in potentially significant and heterogeneous networks provides a new research challenge that demands standard approaches for large-scale distributed optimization, ML, and privacy of acquired data analysis [72,73].

Wearable gadgets, mobile phones, and driverless cars are a few examples of distributed networks that produce a tremendous amount of data each day. Because of the increased processing capability of the devices indicated above, the transmission of private data is a key concern. FL enables privacy to user data by decentralizing data to end devices from central servers and facilitates artificial intelligence (AI) benefits to heterogeneity and sensitive data fields. The FL model is used for two motives: the inaccessibility of adequate data stored on the central server because of the restrictions to data and the protection of private data through edge devices using local data, e.g., clients [19].

Keeping user data privacy delivers probability to influence AI advantages using ML/DL-trained models effectively in different domains. Furthermore, the computational time and power are divided into associated parties instead of the central server by training the iterative model on the end device. Because of the decentralized concept, in the area of ML/DL, FL is the growing field in recent years because of its privacy and security features regarding user data protection [74,75]. Besides privacy, FL also allows ML and DL advantages to other domains where enough data are not accessible for training to create a separate trained ML/DL model. Figure 4 presents the use of federated learning in a VANET environment.

## 4. Blockchain

Blockchain has recently piqued the interest of both academics and industry. It is extremely effective to handle data transferring constraints between heterogeneous devices On the Internet of Vehicles (IOV) while ensuring privacy and security [76]. Blockchain was developed to service banking and digital money (Bitcoin); later on, it was implemented in other spheres of life. ITS has minimized the urbanization transportation problems. The development of wireless sensors in vehicles revolutionizes communication between vehicles, thereby creating Vehicular Ad-hoc Networks (VANETs) [77,78]. VANET improves network traffic flow to provide timely information and facilitate intelligent transportation services. VANET is employed because it has the capability of self-organizing data transmission to on-road vehicles and to allow applications, e.g., safety warnings and assistance in-vehicle driving [79,80]. Instead of cellular vehicular-to-everything (C-V2X), the European Parliament has adopted DSRCs (dedicated short-range communications) [81], because soon, VANET will have broader applications, and it is also ’infrastructure-less’.

Without any central medium, blockchain is a novel technology that permits transactions to different peers in the network and also influences the distributed ledger [82]. To promote data credibility and accountability, several VANET applications have recently been identified as the potential value of this new technology [83]. Because of the autonomous infrastructure of the vehicles, they need to exchange data with more ’trust’ in different intelligent transportation situations, e.g., smart contracts, which already benefit from blockchain technology [84].

In blockchain technology, nodes perform a validation process because they do not trust each other whenever a new block is created in the system [82]. After the validation process, the existing nodes trust the newly created block. The process outlined above is called Proof-of-Work (PoW); certain transactions are valid within the network because of the above process. It is perfect if the newly created block is verified by Full Nodes (FNs) within the network [85], which requires a broadcast of the newly created block to all FNs within the whole network. This process takes several more significant factors from others in blockchain performance in a network, such as total broadcast delay and the number of nodes. The critical factors in blockchain networks include: security and credibility of consensus and blockchain network energy consumption [86].

The main focus of a VANET is to improve credibility and security in data exchange. To improve the security of the VANET data, [87] proposed a three-layer framework. To break the relationship between public keys and real identities from the privacy perspective, [88] proposed an anonymous reputation system based on blockchain. The blockchain was used for modeling data communication between flying drones securely in a 5G network in [89]. For secured storage system and data sharing in VANET, [90] studied a consortium blockchain technique. Another research work also used smart contract technologies and consortium blockchain to accomplish secured data sharing and to store in VANET [91]. The study assumes that edge computing servers in the vehicular network included RSUs suffering from severe privacy and security issues and are not fully trusted. Finally, in electric vehicles, blockchain technology is used to improve the edge computing [92]. According to the perspectives of energy and information interactions between vehicles, context-aware applications are recognized, in which energy involvement amount and data involvement frequency are used to achieve PoW.

Table 1 analyzes and describes the role of blockchain in the Vehicular Internet of Things (VIoT) environment from different features perspectives. Table 1 also illustrates and categorizes existing proposed scenarios.

### Integration of Federated Learning and Blockchain in Vehicular Networks

IoV is capable of handling a large quantity of data transmission, storage, and real-time computations to accommodate the needs of infotainment and safety (roadside) [115]. IoV empowers Vehicle-to-Infrastructure (V2I), Vehicle-to-Vehicle (V2V), and Vehicle-to-Everything (V2X) communication, where V2I is purely dependent on VANET, and RSU is utilized to facilitate authentic and reliable information. On the other hand, V2V is an infrastructure-less communication model mainly deployed during rush hours or in emergencies. Artificial Intelligence, Machine Learning [116], and fuzzy logic [117] methods are adopted to handle vehicles direction, speed, and propagation loss, but these techniques consume large processing power, which is not feasible in the fully distributed architecture [118]. Many vehicle networks require real-time and accurate decision-making within a specified time, which is not suitable for a centralized architecture. FL is the best choice to handle such problems because of its distributed machine learning approach. Mobile devices collect data, which they use for training, and generate local models. After that, local model data are supplied to the aggregator, which takes the average of local models and generates a global model. Mobile devices train each global model until the desired goal is achieved.

The integration of FL and blockchain can solve the challenges in an existing centralized system, such as resource consumption limitations, and is also suitable for the sensitive applications having autonomous vehicles. The decentralized solution will optimize computing resource utilization and provide balanced workloads compared to centralized networks. Furthermore, using cryptography characteristics, data consistency is also ensured in the blockchain. The addition of cryptography characteristics to VANET will offer secure data storage and data transmission within the network. Blockchain saves details of each connected vehicle in VANET as well as traces the newly added vehicles to the network [71]. In VANET, each vehicle’s identity is assured; the process allows data authentication and authorization in VANET devices. Among vehicles, blockchain ensures secure data communication [119], performing the transaction validation and saving locally trained data of different transactions for every vehicle [120].

The data breach is the most challenging issue in critical infrastructures, e.g., military data [121], connected vehicles [122], banking [123], and healthcare [124]. Blockchain provides data encryption techniques to eliminate the risk associated with data breaches and optimize VANET protocols. Furthermore, VANET transfers a large amount of real-time network data to connected vehicles. This process also secures VANET and assures an appropriate data management technique. Providing an FL solution based on the blockchain permits vehicles in VANET to communicate with one another using encrypted identities (IDs) deprived of the central server. The transaction is distributed equally after adding the transaction to the block within all devices in the network [71,125]. Ref. [126] proposed a solution for VANET infrastructures based on the blockchain-FL system. The vehicle’s training model solution is managed by utilizing blockchain and FL technologies.

In VANET, the structure of FL consists of the following two components:

**Centralized Server:** In FL procedure, a centralized server plays a significant part because its main job is to communicate with end devices, such as vehicle clients, to match trained local model updates.

**Vehicle Client:** They have a set of different types of sensors, such as cameras and GPS. Vehicle clients also have computation and communication resources and storage [127]. The vehicle clients train their local ML models with the help of their local data and forward the updated models to a central server after receiving requests from a centralized server.

The following are some of the issues that FL and blockchain integration can solve in VANET:

**Security and privacy:** FL’s primary responsibility is the preservation of data privacy. However, several weaknesses exist in FL from the point of view of privacy and security, such as that the attacker can access confidential information of the end device through their model updates already available on the central server. Furthermore, the malicious end device can also modify the updated local model. The attackers can also launch an injection attack within the central server to send an incorrect model update to attached end devices in the process of the FL algorithm. Because of all of the above reasons, FL security and privacy are most important.

**Scalability:** Local model constraints are significant for FL because they are used in the process of global model updation. Consequently, if the number of devices increases in each learning round, then the performance of FL will also increase. The process of increasing the number of devices in FL is called scalability. Scalability can be accomplished in FL through different methods, such as picking end devices based on computation power and resource optimization.

**Quantization:** The process of reducing local trained model data size during the updating process. With the help of this process, we can increase the throughput time, which will eventually increase the conjunction time of FL.

**Robustness:** The capability of the central system to provide services under attack or the failure of the central server is called the robustness of the central system.

**Sparsification:** The process of the collection of appropriate devices from the available set of devices involved in FL. Throughout the training rounds, only the best suitable set of end devices are allowed in the FL method. Several parameters are involved in selecting the end device, such as unique parameters in available data, size of the selected dataset, computation power, and noisiness in the selected dataset.

## 5. Applications of Integrating Federated Learning and Blockchain in Vehicular Networks

While transmitting data, valuable information can be disclosed through the model parameters by reverse engineering [128]. The disclosure of valuable data motivated researchers and developers to adopt known security and privacy defense methods, e.g., functional encryption and differential privacy, to FL [129].

The following are some of the advantages of FL and BC in the VANET environment:In the modern VANET environment, advanced vehicles have larger battery capacity and are more resourceful than traditional end devices. Using FL and BC as a basic storage and computing unit will improve the competence of VANET.Compared to traditional vehicular networks, for high data transmission, FVN uses heterogeneous communication systems, and for efficient data exchange and update, FVC is introduced.The integration of FL and BC to VANET infrastructure provides the continuous interaction with end devices, but this will also incentivize several entities, such as vehicles, clients, and venues in the FVN participation.As compared to fog- and edge-based networks, FVN is more reliable and secured due to trained data/model offloading to vehicles. The sensitive information is stored in the vehicle’s OBUs in FVN. The training phase is complete without the involvement of third parties.Compared to the traditional fog learning model, FVN provides a well-organized and secure framework from a communication point of view [130].The incorporation of BC into VANET systems will enable data transactions and mitigate malicious activities between several end devices.

Table 2 presents the evaluation of available FL paradigms in the VANET environment.

### 5.1. Federated Learning and Blockchain for Security in Vehicular Networks

The fundamentals of information security CIA must be adhered to by the FL developers and adopters. Many end devices are included in the exposure and training of model characteristics through a decentralized approach, making FL vulnerable to several open risks and attacks. Current research regarding vulnerabilities and frameworks for mitigating risks in the FL technology is limited.

By using BC technology, several researchers and developers made efforts to improve the security of the VANET in recent times. Reference [131] proposed a novel secure spectrum sharing technique for VANET cellular networks based on blockchain; the proposed technique is for VANET and network operators where a new Stackelberg framework is presented for the optimal spectrum approaches. Reference [132] presented BC-as-a-service (BaaS) for IoT devices cohesive with MEC. VANET is used as a base station in the proposed framework, and BC is used for computation-intensive task offloading. For IoT networks based on MEC, [133] proposed a framework for secure data collection. In the proposed framework for authentication, end devices transfer private data to MEC servers. A BC-based decentralized framework is proposed by [134] for the ground to air data sharing in IoT networks. A Cournot framework is designed to achieve maximum advantages from the ground to air sensors. For efficient and secure key distribution and recovery in VANET, [135] proposed an essential distribution technique based on the decentralized group by exploiting mutual healing and private BC protocol. For the security of VANET, several frameworks are proposed by different researchers based on the BC technique; some of the proposed frameworks are based on the BC network implementation under the FL framework for the applications of MCS. Reference [136] proposed a novel technique of privacy-preserving and secure FL for VANET. The author also presented a decentralized FL framework based on the BC techniques focused on user privacy to protect data contribution verification and data training between VANET. Table 3 provides solutions to FL limitations.

### 5.2. Federated Learning and Blockchain for Privacy Preservation in Vehicular Networks

Shortcomings of VANET are privacy, availability, integrity, identification, and confidentiality prevention from incoming attack [137,138]. Authentication of each vehicle in a network is a key security feature that must be ensured while spreading data within or across the network. Previously, the identification system was based on Public Key Infrastructure (PKI), where each vehicle in a system exchanges its private encrypted identification message to the Local Authentication Center (LAC), which takes enough time to identify a single vehicle. Periodic encryption and decryption create overhead problems in a network, which in return affect the efficiency and reliability of a network [139].

Privacy assurance is becoming the primary concern as technology is intervening in our daily life [140]. Due to the limitation of mobility and resources of vehicles in VANET, there are two main problems with deploying special data privacy system [141,142]. To ensure vehicles’ data privacy and reduce latency, FL enables several entities with fewer resources, e.g., RDUs and vehicles, to combine and train a general model using local data of devices. During the data transformation process, to preserve data privacy, the raw data of the network is distorted by plotting this in different models with less sensitive information [143]. Leveraging FL, the integration of two different components can mitigate data privacy problems in VANET [144], as can be seen in Figure 5.

To address the privacy issues in VANET, various researchers tried to solve the issues from various research angles. Existing privacy frameworks: the differential privacy framework [145], its extensions [146], and the classic privacy framework are not sufficient to solve the privacy issues in VANET. To date, researchers did not find an optimal global solution for data privacy, and utility protection [147]. Ref. [148] proposed a encryption-based technique to solve the privacy issues. The proposed technique is helpful to a satisfactory level, but the proposed technique does not help in big data situations. FL can accomplish effective communication by transferring updated data between global models and local models [149,150]. FL solves the privacy issues to the maximum level, but another issue can arise: if the central model is poisoned or compromised, the adversaries may launch successful attacks. To solve the trust issues in FL, blockchain technology is introduced in such situations [151]. By forcefully incorporating privacy solutions through blockchain, it will decrease the efficiency of the application. FL overcame several data privacy issues by dividing the data into two parts: global aggregation and local training in the learning phase, but several other security issues arose. Table 4 presents the literature review of security, privacy, and energy efficiency in the VANET environment.

## 6. Real-World Use Cases/Project

The below case study describes a threat scenario in the VANET environment. For MEC and edge caching, deployed entities such as vehicles and RSUs are fully furnished with computing resources and caching in the described scenario. The adversary in the scenario aims to access the vehicles and RSUs sensitive data illegally through eavesdropping by malware implanting, by initiating illegal sensitive data transmission, or by exploiting the vulnerabilities of the systems to reach the targeted RSUs or vehicles in the network to access devices private data without their knowledge. During the transmission of data from vehicle X to Z, there is the possibility that the adversaries may access private data such as in-car data transmission, V2X data transmission, and during the transmission of data between users. The deployed vehicles’ inadequate storage and computing resources motivate them to transfer their data through V2V and V2R communications. During the data transmission process, there is also a chance that vehicles may unintentionally transfer their private data with other vehicles. During data sharing, the vehicle’s private data may be hidden in standard data transmission, and this may cause severe physical damage.

If we summarize the above scenario, the vehicular data leakage threats can be categorized into two parts: the data transferring from vehicles called uplink and the data caching from RSUs called the downlink. The following two challenges must be solved to mitigate data leakage in the VANET environment.

The process of raw data to save from leakage in other applications, such as data allocation and sharing, deprived of breaching the data usability, and only on the requests of other vehicles will the newly created sensitive data be shared.Without any precise tasks, the existing unusable data must be cached, and from the perspective of specific attack types detection of data, leakage must be identified. The data leakage may be because of unknown system vulnerabilities or unintentional behavior.

### Ongoing Projects in Blockchain and VANET

In the past few years, the European Commission (EC) actively set new pilot projects, proof of concepts, and European Union (EU) inventiveness to test, explore, and understand legal requirements, policy, regulations, research, and funding required for technologies such as blockchain and distributed ledger. The European commission’s current projects are grouped into different technologies and topics, such as education, decentralized data management, e-identity, healthcare, privacy, IoT, cybersecurity, music and media, smart homes, smart cities and grids, industrial technologies, circular economy, and environment. This list may be expanded upon in upcoming funded projects in several other blockchain technology areas. Figure 6 presents the applications of blockchain technology in different ongoing projects.

## 7. Research Challenges, Open Issues, and Future Directions

### 7.1. Privacy and Security Issues

VANETs are prone to remote attacks that can lead to vehicles malfunctioning. Some examples are attackers gaining access to the vehicles, leading to steering control, down the engines, applying brakes, etc. Even though FL and blockchain can control privacy leakages, the attackers can reconstruct the attributes from the model updates sent to the central server from the individual vehicles (poison attacks). To address the privacy and security issues, advanced encryption algorithms can be used to encrypt the model updates that are sent from the individual devices to the cloud and also when the vehicles are communicating with each other [166].

### 7.2. Quality of the Data

The federated learning (FL) model primarily depends on the data quality, since they are supposed to make efficient decisions. Automated vehicles are embedded with sensors that collect data from almost all vehicle components. These data have to be annotated and pre-processed well so that vehicles can make efficient decisions in a real-time environment. Low-quality annotations might cause a vehicle to misinterpret what is happening on the road and end up in a fatal accident. Semi-annotated datasets can be used in which first the dataset are annotated automatically and then verified by humans manually. Secondly, state-of-the-art data pre-processing can be done at the hardware and model levels. At the hardware level, smart sensors (reprogrammable) can be introduced. Since sensors have low computational power, they can perform minor but effective pre-processing tasks. Pre-processing can be deployed using query-based data processing to filter out redundant data. These sensors are not easily available in the market. However, there is some work going on to build them [167]. At the model level, noise can be removed, and missing data can be imputed. Another case of such misinterpretation can be a small dataset from a particular node (i.e., sensor or vehicle in a broad connected network), which results in biasness in the FL model. Considering the overall connected and automated vehicle network, communicating among each vehicle in the network should produce feasible data so that vehicles for which one local model is used can participate fully in the FL model training process.

### 7.3. Lack of Interpretability/Justification

FL models’ fairness and interpretability/justification are critical to making humans understand why the learning model has made a decision and improving the learning process. The higher the interpretability, the easier it is to understand why certain decisions or predictions have been made. Most models are considered black-box since they only make a decision; however, they do not interpret them. Therefore, this issue can be handled by converting the black-box model to a white box. White box models are interpretable and justify how a decision is made. One such example of a white-box model is the decision tree (DT), a rule-based approach. Features are split based on information gain from root to leaf nodes, where the leaf nodes are the labels or classes to predict.

### 7.4. Near Real-Time Decisions

Controlling and managing network connections between cars and infrastructure is a significant task. Vehicles that are connected should maintain a constant and highly reliable connection with fog devices to prevent communication system transmission failures and make decisions in time. In automotive networks, intermittent connections owing to vehicle movement or excessive packet loss must be avoided.

### 7.5. Generation of Class Labels in Real Time

Labels are class affiliations that are specified with every feature set. During training, labels are supplied; however, at test time, labels are hypothesized. Selecting proper class labels in a real-time environment is vital, since picking erroneous labels would lead to mass catastrophe. Older technology using machine learning is still experiencing this difficulty, and exact measurements are needed to solve such problems in an extremely important environment.

### 7.6. Handling Big Data

Traditional centralized VANET technology is ineffective in dealing with vast volumes of traffic (data) provided by smart cars, such as video and sensor data. More servers are necessary for traffic information in scattered locations to gather and analyze such enormous data in real-time. Cloud-based VANETs might be a viable option. Generally speaking, uploading to vehicles utilizes data downloaded from the cloud. With time and effort, as the number of cars on the roads grows, cities’ current cloud computing paradigm can barely meet the location awareness standards, latency, and mobility assistance.

### 7.7. Enabled Network Intelligence

In the future, cars will be equipped with many sensors, and the edge cloud will gather data and preprocess it before sharing it with other portions of the network. In such a network, incoming data packets are classified based on their importance, e.g., time-critical events and non-real-time events. Advanced Federated Learning and Blockchain algorithms must preprocess the incoming data and efficiently spread it accordingly.

**Scalability:** The implementation of blockchain and federating learning in the VANET environment produce different scalability issues. Blockchain is a decentralized technology [168,169]. Therefore, its applications should be implemented on the underlying medium. Blockchain and federating learning cannot be implemented if the scalability and performance of the systems are not significant and sufficient. Based on decentralization and data security, the blockchain scalability issues are categorized into the following different aspects: performance limitations, network delays, and consistency. For ensuring blockchain security, end nodes required transaction data consensus [170].

**Collaboration between Off-Chain and On-Chain Data Storage:** The two main ways of data storage, blockchain and traditional information-based, are used in VIoT [171]. The performance of blockchain can be improved with the help of computing systems and off-chain storage. On the other hand, blockchain systems confirm the data’s credibility and safe sharing from traditional information systems [172]. Although all of the above required combining traditional information methods and blockchain technology, data consistency and relevancy are the most critical points from the off-chain and data on-chain perspectives [173].

## 8. Conclusions

The era of Smart Transport Infrastructure heavily relies on IoT. The Internet of things revitalizes the world with tremendous capabilities and potential to be exploited in-vehicle networks. Advanced ML techniques are in use to further expand the smartness of the Smart Transport Infrastructure. Security is the ultimate need of such real-time systems, where a small failure or problem may cost lives. Intelligence transportation infrastructure (ITI) systems require all three pillars of security to be taken care of, including confidentiality, integrity, and availability. The critical point of entry to the ITI is via its network. There are three primary domains of the ITI, which are Ad-Hoc Network Domain (AHND), Roadside Network Domain (RSND), and In-Vehicle Network Domain (IVND) (IVND). In this survey, we have explored the integration of FL and blockchain in the VANET environment from the privacy and security viewpoints and the applications of FL and blockchain in VANET. We also reviewed current constraints and potential research directions in integrating FL with blockchain. 

## Figures and Tables

**Figure 1 sensors-22-04394-f001:**
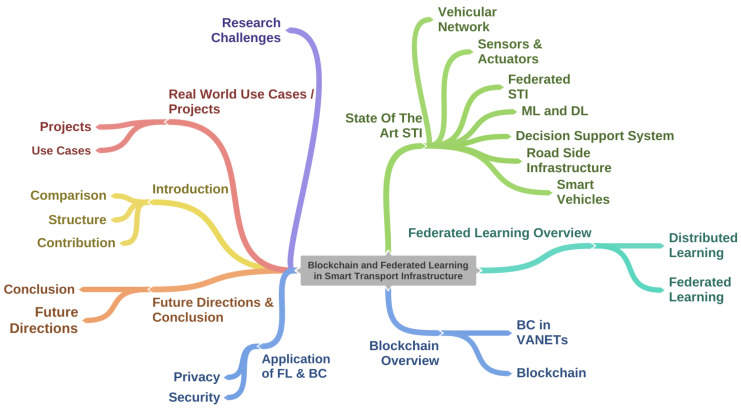
Structure of the proposed survey.

**Figure 2 sensors-22-04394-f002:**
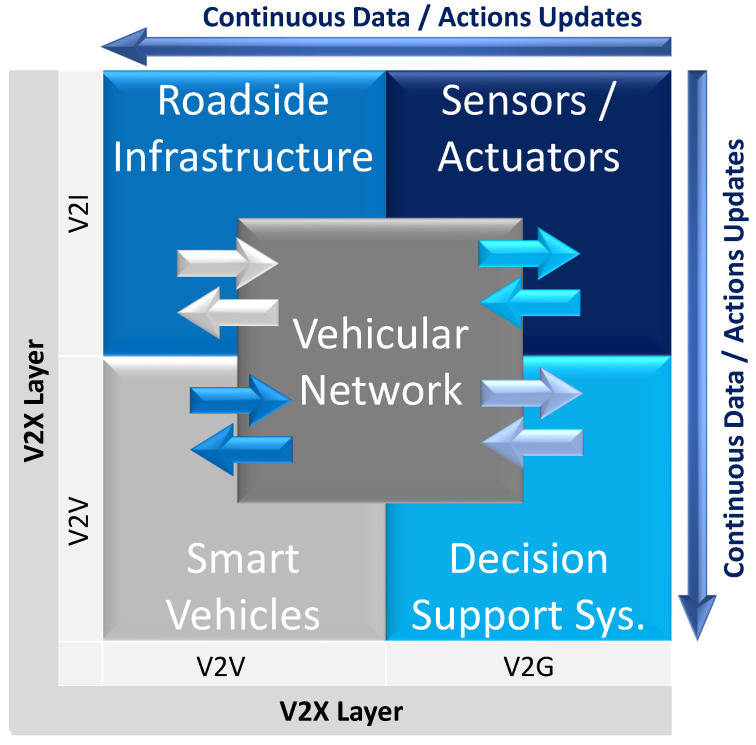
Smart transport infrastructure.

**Figure 3 sensors-22-04394-f003:**
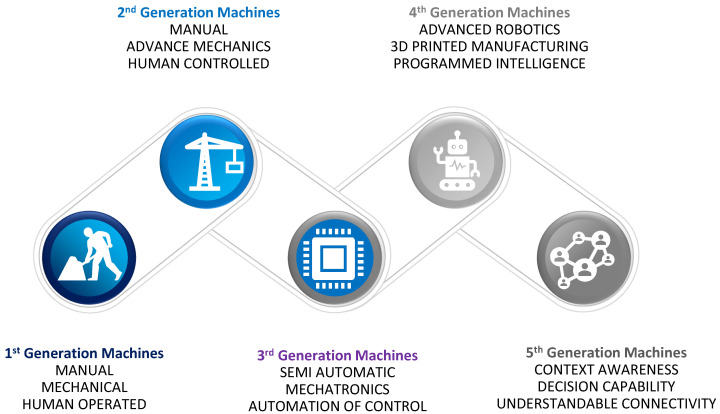
Machine technology evolution.

**Figure 4 sensors-22-04394-f004:**
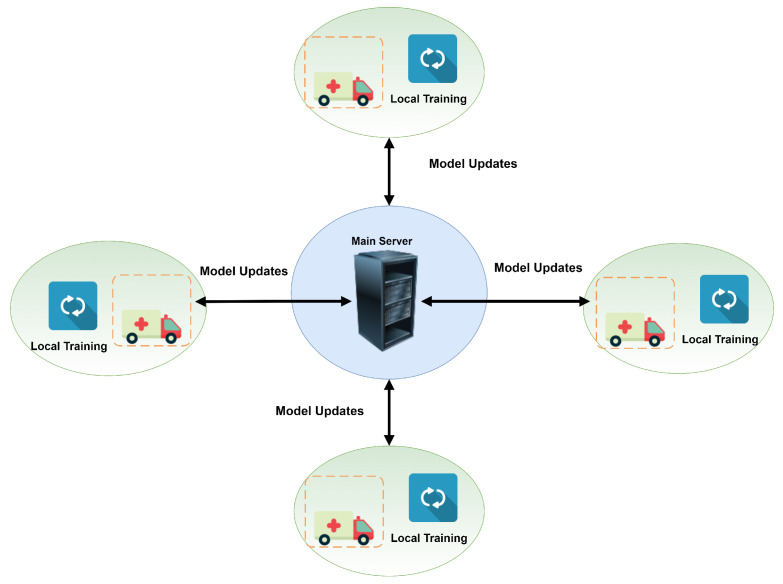
Usage of federated learning in a VANET environment.

**Figure 5 sensors-22-04394-f005:**
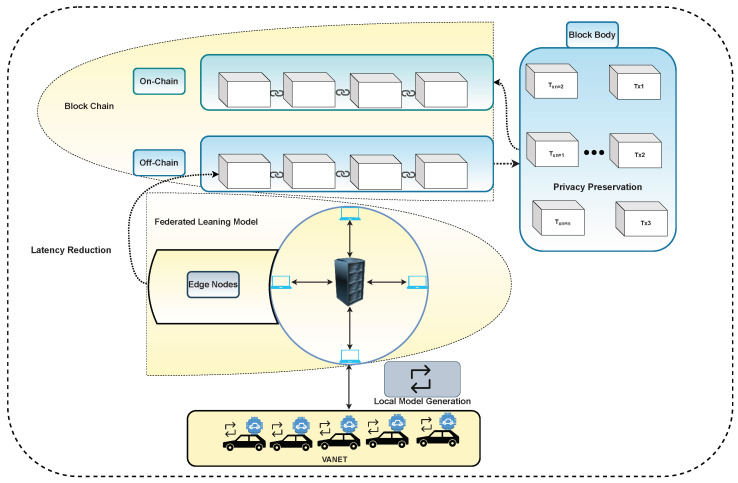
Integration of FL and blockchain in the VANET environment.

**Figure 6 sensors-22-04394-f006:**
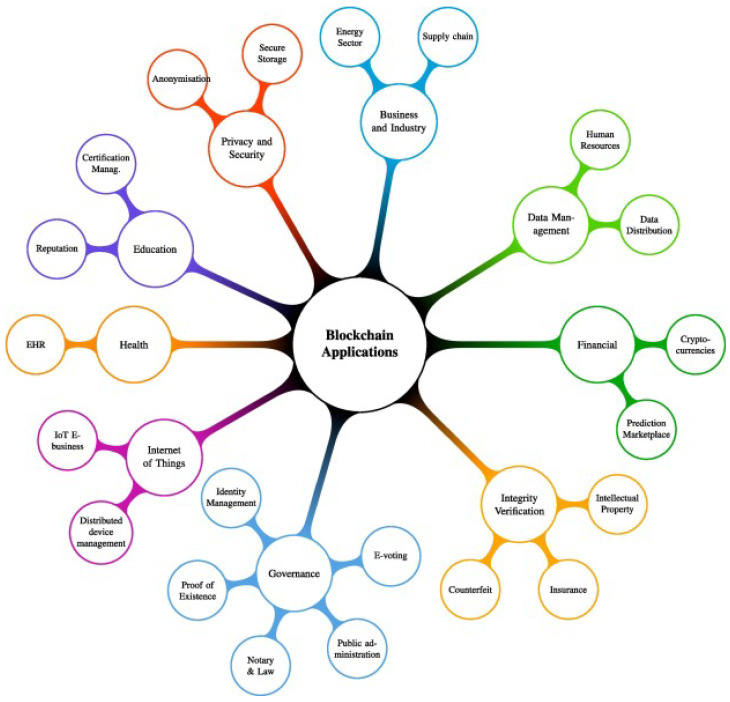
Applications of blockchain in ongoing projects.

**Table 1 sensors-22-04394-t001:** Integration of vehicular internet of things and blockchain.

Ref	Categories	Research Contribution
[93]	Vehicular IoT architecture	Author proposed SDN architecture of fog and 5G systems based on blockchain.
[94]		Author proposed SD-VANET framework based on blockchain.
[95]	Privacy-preservation in VIoT	Authentication system based on blockchain and privacy-preserving.
[96]		Author proposed hybrid blockchain privacy framework for VIOT.
[97]	Data monetization in VIoT	Resource trading framework based on blockchain.
[98]		Data trading method based on Consortium blockchain.
[99]		Loaning system and data trading based on blockchain.
[100]	Data management in VIoT	Author proposed DQDA mechanism based on blockchain for VIoT.
[101]		Author proposed mobile crowd sensing method with blockchain for data management in VIoT.
[102]		Block verification and miner selection solutions for VIoT.
[76]		Author proposed hierarchical blockchain resource scheduling which is most suitable for blockchain-enabled Internet of Vehicles and data exchange demands
[103]	Certificate management in VIoT	For traffic signal author proposed a semi-centralized control mode.
[104]		For anonymous reputation, the author proposed a blockchain-based system.
[105]		For the VIoT system, the author proposed a decentralized key distribution and management technique.
[106]		Privacy-preserving authentication technique based on blockchain.
[107]	Trust management in VIoT	Trust management and privacy-preserving framework based on blockchain.
[108]		Author proposed a novel scheme for anonymous cloaking region.
[109]		Author proposed a novel protocol called Vehicular announcement.
[110]	VIoT security	A framework for traffic event authentication based on blockchain.
[111]		For the VANET environment, the author proposed distribute trust mechanism.
[112]		A novel mechanism for trust clustering for VANET environment.
[113]		Author proposed a new security technique called intelligent vehicle trust point for VANET environment.
[77]		Author proposed the certificate less message technique to ensure non-repudiation and anonymity for traffic-related message reporters. Multiple participants are required to validate the authenticity of information.
[114]		Blockchain-based APP for VIoT security.

**Table 2 sensors-22-04394-t002:** Evaluation of FL paradigms in the VANET environment.

Characteristics	Traditional FL	FCN Private Data	FVN Combined
Flexibility	Easy to deploy	Easy to launch	Harder to deploy
Processing	Data parallel	Data parallel	Model parallel + data parallel
Comp. unit	Mobile device	Vehicle	FVC
Computation	Limited	Medium	High

**Table 3 sensors-22-04394-t003:** Blockchain-enabled federated learning.

Limitation of FL	Solution Provided by Blockchain-Enabled FL
FL is not suitable for the aggregating updates while selecting vehicles and maintain GM.	Blockchain provides a solution to all these problems through its decentralized storage and further maintaining the FL model. Blockchain can be used to store GM.
High speed is required for the server to gather information and update vehicles (clients).	
Express bandwidth is required.	
Skewing in GM can also be expected because of biasness.	
FL cannot detect the internal attacks by malicious node while updates are gathered from every vehicle in a network causing GM unable to link up.	

**Table 4 sensors-22-04394-t004:** Literature review of security, privacy, and energy efficiency.

Ref.	Contribution	Environment	Focused Area
[152]	In this research work, the author proposed a privacy reserving communication scheme based on VANET. The proposed framework meets the contextual and content privacy requirements. It used identity-based encryption and an elliptic curve cryptography scheme.	ITS	Security and Privacy
[153]	In this research work, the author proposed a contest-aware quantification technique to overcome security issues in VANET based on the Markov chain method.	VANET	Security
[154]	Based on wireless communication, the author presents a literature review of existing work related to VANET technology. The author also presents research directions and open issues for the integration of SDN with VANET.	SDN, IoT	Security
[155]	The proposed work addressed different privacy and security issues regarding VANET. The paper also presents the solutions to privacy and security issues.	VANET	Security and Privacy
[156]	The proposed work presented an overview of secure and smart communications using the IoT-based VANET technique to overcome traffic congestions in CPS, known as networks of IoV.	CPS, IoV	Security
[157]	The author made different clusters of vehicle packets of the specific cellular tower in an IoT environment. This process simplified communication, and VANET architecture reduces energy consumption and network delays.	IoT	Energy efficiency
[158]	The author proposed a lightweight end-to-end security solution for SDNV. The proposed objectives are achieved on two-level: RSU-based authentication technique and personal IDS. The lightweight security solution will also provide privacy.	SDNV	Energy
[159]	The author proposed a source location privacy preservation method based on smart energy for sustainable city roads. The proposed technique hides source location based on acceleration, distance, speed, and trust.	IoT	Energy and Privacy
[160]	The author proposed a new algorithm for multi-hop transmission called fuzzy clustering routing. The author also analyzed clustering limitations, which are performed through different algorithms. To transfer data, multi-hop routing was used.	IoT	Energy
[161]	This paper presented the different notions of blockchain and its usability in IoT networks. The author presented different privacy issues regarding the implementation of blockchain in IoT. The author presented FL usability in IoT networks, privacy risks, and taxonomy.	IoT	Privacy
[162]	Among different elements elaborate to manage a group of vehicles containing data, the author proposed a blockchain framework. The author integrates VPKI for blockchain to provide privacy and membership association.	VANET	Privacy
[163]	The author presented the fundamentals of IoT and blockchain. Then, the author presented a comprehensive literature review based on blockchain techniques for VIoT through the technical issues and problems. At the end of the paper, the authors present the future research direction regarding VIoT and blockchain.	VIoT	Energy and Privacy
[164]	The proposed research work analyzed and described existing supply chain, healthcare, VANET, and IoT access control through blockchain security methods. The author also presents a comprehensive survey regarding blockchain security.	IoT	Security and Privacy
[165]	The author proposed a new technique called FL-Block (blockchain FL) to overcome the existing issues in FL privacy. The local learning update is transferred to global learning using blockchain through this technique.	Fog computing	Privacy

## Data Availability

Not applicable.

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
