# Peer review of "Integration of Blockchain Technology and Federated Learning in Vehicular (IoT) Networks: A Comprehensive Survey"

_sensors, 2022, doi:10.3390/s22124394_

Round 1
Reviewer 1 Report
This paper surveys the vehicular network with Federated Learning and Blockchain. This work is meaningful and timely. But, there are some issues given as follows:
(1) The authors are suggested to illustrate the Federated Learning more clearly by presenting an operation diagram of this technique.
(2) The authors are suggested to illustrate the Blockchain operation more clearly by presenting a diagram of this technique.
(3) The authors are suggested to illustrate the integration of Federated Learning with Blockchain by presenting a diagram to show the main operation prcedures.
(4) On page 6, the sentence "Decision support systems act as the brain of the STI. They gather data from diverse sources, including vehicles, roadside sensors, roadside monitors, intelligent cameras, and weather sensors." appears twice in different paragraphs. Please remove one.
(5) The two sections about research challenges and future directions seem not sufficient enough. The authors are suggested to extend these two sections more sufficiently and more logically.
(6) In the last section, it is not necessary to outline the future work, as this part should belong to the section of furture directions.
Author Response
This paper surveys the vehicular network with Federated Learning and Blockchain. This work is meaningful and timely. But, there are some issues given as follows:
(1) The authors are suggested to illustrate the Federated Learning more clearly by presenting an operation diagram of this technique.
Response: Many Thanks for your Valuable Comment, we have added Figure 4 in the manuscript.
(2) The authors are suggested to illustrate the Blockchain operation more clearly by presenting a diagram of this technique.
Response: Many thanks for your Valuable, we have added Figure 5 in the Manuscript.
(3) The authors are suggested to illustrate the integration of Federated Learning with Blockchain by presenting a diagram to show the main operation prcedures.
Response: Many thanks. We can confirmt that we have now FIgure 8 showing the in illustrate the integration of Federated Learning with Blockchain.
(4) On page 6, the sentence "Decision support systems act as the brain of the STI. They gather data from diverse sources, including vehicles, roadside sensors, roadside monitors, intelligent cameras, and weather sensors." appears twice in different paragraphs. Please remove one.
Response: Many Thanks for your Valuable comment, we have removed the duplicated sentences.
(5) The two sections about research challenges and future directions seem not sufficient enough. The authors are suggested to extend these two sections more sufficiently and more logically.
Response: Many thanks for the comment. We can confirm that we have extended and improved these two sections.
(6) In the last section, it is not necessary to outline the future work, as this part should belong to the section of furture directions.
Response: Many Thanks for your Valuable comment, we have removed separate section of future work and marge it with the conclusion and future work section.
Reviewer 2 Report
The paper presents a comprehensive survey for integration of blockchain technology and federated learning in vehicular (IoT) Networks. According to the Authors, the elaborated survey explores the vehicular network and STI in detail and sheds light on the blockchain and FL with real-world implementations. The Authors tried to present the current research challenges and future research directions related to integrating FL and blockchain for vehicular networks. The topic is interesting and the paper is well corresponding to the journal aim and scope.
The paper is well structured. The Authors highlighted their contribution as well as the conducted survey structure. However, in Conclusions section, it is worth referring to the contribution presented in the Introduction.
Overall, it is an interesting review paper. The question is whether it should be classified as an article or as a review?
Author Response
The paper presents a comprehensive survey for integration of blockchain technology and federated learning in vehicular (IoT) Networks. According to the Authors, the elaborated survey explores the vehicular network and STI in detail and sheds light on the blockchain and FL with real-world implementations. The Authors tried to present the current research challenges and future research directions related to integrating FL and blockchain for vehicular networks. The topic is interesting and the paper is well corresponding to the journal aim and scope. The paper is well structured. The Authors highlighted their contribution as well as the conducted survey structure. However, in Conclusions section, it is worth referring to the contribution presented in the Introduction.
Overall, it is an interesting review paper. The question is whether it should be classified as an article or as a review?
Response: Thansk for the possitve comment. It is a review article.
Round 2
Reviewer 1 Report
The reviewer's concerns have been addressed satisfactorily. The reviewer has no further comments.
Author Response
Thank you for your positive feedback